# Correlation between Neurotransmitters (Dopamine, Epinephrine, Norepinephrine, Serotonin), Prognostic Nutritional Index, Glasgow Prognostic Score, Systemic Inflammatory Response Markers, and TNM Staging in a Cohort of Colorectal Neuroendocrine Tumor Patients

**DOI:** 10.3390/ijms25136977

**Published:** 2024-06-26

**Authors:** Radu Cristian Cîmpeanu, Mihail Virgil Boldeanu, Roxana-Viorela Ahrițculesei, Alina Elena Ciobanu, Anda-Mihaela Cristescu, Dragoș Forțofoiu, Isabela Siloși, Daniel-Nicolae Pirici, Sergiu-Marian Cazacu, Lidia Boldeanu, Cristin Constantin Vere

**Affiliations:** 1Doctoral School, University of Medicine and Pharmacy of Craiova, 200349 Craiova, Romania; cimpeanu_r@yahoo.com (R.C.C.); roxana.blendea@gmail.com (R.-V.A.); elena.ciobanu210@gmail.com (A.E.C.); anda_enculescu@yahoo.com (A.-M.C.); fortofoiudragos@gmail.com (D.F.); 2Department of Immunology, Faculty of Medicine, University of Medicine and Pharmacy of Craiova, 200349 Craiova, Romania; isabela_silosi@yahoo.com; 3Department of Histopathology, Faculty of Medicine, University of Medicine and Pharmacy of Craiova, 200349 Craiova, Romania; danielpirici@yahoo.com; 4Department of Gastroenterology, University of Medicine and Pharmacy of Craiova, 200349 Craiova, Romania; cazacu2sergiu@yahoo.com (S.-M.C.); vere_cristin@yahoo.com (C.C.V.); 5Department of Microbiology, Faculty of Medicine, University of Medicine and Pharmacy of Craiova, 200349 Craiova, Romania; lidia.boldeanu@umfcv.ro

**Keywords:** neurotransmitters, Prognostic Nutritional Index, Glasgow Prognostic Score, systemic inflammatory response markers, colorectal neuroendocrine tumors

## Abstract

Neuroendocrine tumors are uncommon in the gastrointestinal system but can develop in the majority of the body’s epithelial organs. Our goal was to examine the presence and clinical application of serum dopamine (DA), serotonin (ST), norepinephrine (NE), and epinephrine (EPI), in addition to determining the significance of the Prognostic Nutritional Index (PNI), Glasgow Prognostic Score (GPS), and systemic inflammatory response (SIR) markers as a prognostic factor for patients with colorectal neuroendocrine tumors (CR-NETs), in various tumor–node–metastasis (TNM) stages. We also wanted to identify the possible connection between them. This study included 25 consecutive patients who were diagnosed with CR-NETs and a control group consisting of 60 patients with newly diagnosed colorectal cancer (CRC). We used the Enzyme-Linked Immunosorbent Assay (ELISA) technique. This study revealed that CR-NET patients showed significantly higher serum levels of DA compared to CRC patients. We showed that serum DA was present in the early stages of CR-NETs, with increasing levels as we advanced through the TNM stages. Moreover, we found a close relationship between the levels of DA and the inflammation and nutritional status of the CR-NET patients in this study. CR-NET patients from the PNI < 47.00 subgroup had a higher level of DA than those from the PNI ≥ 47.00 subgroup. Pearson’s correlation analysis revealed correlations between DA, PNI, and the neutrophil/lymphocyte ratio (NLR) and the platelet/lymphocyte ratio (PLR). Both hematological indices were negatively correlated with albumin (ALB). Our investigation’s findings relating to the PNI, GPS, SIR, and DA indicate that these tools can be markers of nutritional and systemic inflammatory status, are simple to use, and are repeatable. Further research on this topic could provide valuable insights into which biomarkers to incorporate into clinical practice for the management of CR-NET patients.

## 1. Introduction

Neuroendocrine tumors (NETs) are a group of neoplasms that originate from cells produced from the neural crest found all over the body. At the neuronal and endocrine junction, neuroendocrine cells produce hormones that are reflected in the appearance and natural history of NETs [1]. Although current epidemiologic data are lacking, it is believed that the incidence and prevalence of NETs are increasing [2].

The paucity of symptoms in the early stages, the frequency of non-specific gastrointestinal symptoms, and the absence of particular tumor markers have made it difficult to diagnose NETs, even with advances in our understanding of the molecular biology of the disease. Patients who do not receive a diagnosis in a timely manner frequently present with advanced disease and a bad prognosis.

The brain–gut axis is a complex communication pathway that plays a key role in the formation and carcinogenesis of tumors in the central nervous system (CNS), enteric nervous system, and endocrine-immune system. Information flow between the gut and brain is accomplished via the widely dispersed neuropeptides and neurotransmitters. An increasing body of research indicates that neurotransmitters and neuropeptides might influence the growth, migration, invasion, and angiogenesis of tumors [3,4]. Numerous investigations have verified that norepinephrine/noradrenaline (NE) and epinephrine/adrenaline (EPI) stimulate angiogenesis, development, invasion, and activity in numerous stress-induced malignancies by upregulating the expression of certain essential proteins linked to carcinogenesis [5,6,7]. Dopamine (DA), in contrast to EPI and NE, can influence carcinogenesis in either a positive or negative way [8]. Inflammasomes in cancer and immune effector cells are two ways that peripheral DA influences immunomodulation and aids in the development of tumors [9]. Serotonin (ST) has been linked to tumor biology through recent evidence as a regulator of proliferation, regeneration, and repair [10]. Serotonin levels in foregut and midgut carcinoids are typically higher than those in hindgut carcinoids [11]. For many different tumor cell types, including those of the bladder, pancreatic, lungs, and particularly colon cancer, ST may act as a mitogen. ST functions as a mucosal signaling molecule in addition to a neurotransmitter [1,8].

The degree of differentiation, primary tumor size, tumor grade, and stage are all known prognostic variables in NET [12,13,14,15]. Although they are still being investigated, additional variables like advanced age at diagnosis, pancreatic tumor location, the occurrence of synchronous metastases, and the functional nature of the tumor have been linked to a worse prognosis [16,17,18,19].

Improved understanding of prognostic factors seems to be crucial in assisting the doctor in selecting a therapeutic approach that is tailored to the severity of the illness.

The metastatic process and clinical stage are associated with the course and prognosis of colorectal cancer (CRC). Currently, the most important staging indicator is tumor–node–metastasis (TNM) staging. The limitations of TNM staging arise from the possibility of disparate clinical outcomes among patients at the same stage. Consequently, a number of indicators are required in order to collaboratively evaluate the patient’s CRC status and provide appropriate treatment [20].

Numerous laboratory biomarkers have been investigated in relation to nutritional status or systemic inflammation. Recent research has revealed that immunological and nutritional factors, such as the Prognostic Nutritional Index (PNI), Glasgow Prognostic Score (GPS), and systemic inflammatory response (SIR) markers, can predict the prognosis of cancer [20,21]. PNI is determined by measuring the absolute lymphocyte (LYM) count and serum albumin (ALB) level. Serum albumin level and C-reactive protein (CRP) are the two factors that GPS is based on. Patients’ nutritional and immunological status is reflected by PNI and GPS, while their immune condition is reflected by SIR [22,23,24,25,26,27].

The relationship between inflammation and tumors has been the subject of a great deal of research in recent years. Combinations of these SIR markers, such as the lymphocyte/monocyte ratio (LMR), platelet/lymphocyte ratio (PLR), and neutrophil/lymphocyte ratio (NLR), are indicators of active tumor inflammation and are crucial in accelerating the progression of tumors [28,29,30,31]. A higher level of NLR, PLR, and LMR is typically linked to a worse prognosis for tumor patients [32,33,34,35,36,37,38]. More recently, research [39,40,41] has demonstrated that the composite inflammatory markers of NLR, PLR, and LMR can also be utilized as prognostic indicators for patients with CRC. In gastroenteropancreatic neuroendocrine neoplasms (GEP-NEN), several previous studies have been conducted, including those involving NLR and PLR as prognostic factors, particularly in pancreatic neuroendocrine tumors (Pan-NETs) [35,42,43,44,45,46,47].

Even though inflammation and cancer are closely related, further research is needed to determine the processes underlying the increased NLR, PLR, and LMR in patients with poor prognoses.

In choosing the research theme, we started from the finding that we did not find any study that addressed the evaluation of the presence of DA, ST, NE, and EPI in the serum of patients with colorectal neuroendocrine tumors (CR-NETs), the investigation of the involvement of PNI, GPS, and SIR markers as a prognostic factor, and the use of the combination of PNI, GPS, and SIR markers as an additional index based on the current TNM stages system.

In order to establish a specific reference for the prognosis of CR-NETs, our goal was to examine the presence and clinical application of serum DA, ST, NE, and EPI, in addition to determining the significance of PNI, GPS, and SIR markers as a prognostic factor for patients with CR-NETs in various TNM stages. We also wanted to identify the possible connection between them.

## 2. Results

### 2.1. Patient Demographic and Clinical Characteristics

This study included 25 diagnosed patients with CR-NETs aged between 42 and 82 years, with a mean ± standard deviation (SD) age of 64.52 ± 11.13, including 11 females and 14 males, and a control group of 60 diagnosed patients with CRC with a similar age (between 45 and 84 years, with a mean ± SD of 68.75 ± 8.46) and gender distribution (29 females and 31 males); there were no statistically significant differences in age and gender (*p* ≥ 0.05).

In Table 1 are indicated the patient demographic and clinical characteristics. Looking at the area of residence, we found that more patients with CR-NETs were diagnosed in the urban area, with significant differences (rural/urban, 10/15 patients, *p* < 0.05).

From the point of view of the TNM stage, in our study, no CR-NETs were identified with TNM stage IV. Depending on the tumor grade, there were 14 patients (56.00%) defined as G1 NET (well differentiated) and 11 (44.00%) as G2 NET (moderately differentiated); in our study, no poorly differentiated tumors were identified. Analyzing the anatomical location of the CR-NETs, we noticed that no tumors were identified in the ascending colon and rectal segments.

### 2.2. Comparison of the Neurotransmitters, PNI, GPS, and SIR Markers in CR-NET and CRC Groups

Our study highlighted that there were statistically significant differences between the CR-NET and CRC groups, regarding the neurotransmitters (mean ± SD): DA (871.53 ± 411.93 vs. 796.09 ± 588.48 pg/mL, *p* = 0.043), and ST values (724.62 ± 396.01 vs. 477.36 ± 358.60 ng/mL, *p* = 0.021) (Table 2).

Regarding the variation of the biological parameters in the study group of patients, statistically significant variations were recorded among the WBC (9.91 ± 2.24 vs. 8.60 ± 1.96, *p* = 0.046), NEU (6.30 ± 1.84 vs. 5.77 ± 1.74, *p* = 0.048), and LYM (2.82 ± 0.67 vs. 1.98 ± 0.79, *p* = 0.039).

Comparing the inflammatory indices in the CR-NET and CRC groups, a statistically significant higher value was found among the PLR (136.77 ± 59.76 vs. 165.80 ± 82.24, *p* = 0.031) and LMR (4.97 ± 2.28 vs. 3.73 ± 1.86, *p* = 0.045).

The median NLR, PLR, and LMR were 2.58, 118.67, and 4.31, respectively, and were chosen as the cutoff in the CR-NET group. In the CRC group, we determined the following medians: 2.95, 148.38, and 3.48, respectively. Provided that 47.00 was the median value among the 25 CR-NET patients and 48.00 for the 60 CRC patients, we used the median of PNI scores as classified criteria that were divided into two groups: low-PNI (<47.00 and <48.00, respectively) group and high-PNI (≥47.00 and ≥48.00, respectively) group.

There were statistically significant differences between the serum concentrations of CRP (40.00 vs. 15.50 mg/dL, *p* = 0.027), and there was no difference in ALB levels (4.70 vs. 4.80 g/dL, *p* = 0.392). This finding may suggest that patients with CR-NETs have a considerably higher inflammatory status compared to those with CRC, as also evidenced by the higher number of LYMs; it is well recognized that LYMs are crucial to the host’s immune response to prevent the growth and spread of tumors [48]. Patients in our study group also presented considerably higher serum values of ST as shown above. Hanoun et al. [49] and Yoo et al. [50] showed that ST regulates inflammation by affecting the immune system. The role of ST in the gastrointestinal inflammatory response by activating immune cells to release inflammatory cytokines has been more clearly demonstrated in a number of animal studies [51,52,53]. Other studies have highlighted the presence of bidirectional neuroimmune interactions at the gut level in the regulation and consequences of intestinal inflammation, such as the central role that serotonin plays as a signaling molecule in triggering, intensifying, and combating inflammation [54,55]. These reports validate the gut’s neuroimmune connections. The above may represent possible explanations for why patients with CR-NETs showed significantly higher values of the CRP and inflammatory status compared to CRC patients.

According to the GPS value, which reflects both the inflammatory and nutritional status, calculated using ALB and CRP levels, we observed the same particularity in both groups that most patients (16/25 CR-NET patients and 39/60 CRC patients, respectively) had mild to moderate malnutrition.

### 2.3. Comparing the PNI and GPS Groups’ Clinical Features between the Study Groups

Based on a PNI cutoff value of 47.0 and 48.0, respectively, we divided both groups into two subgroups: low-PNI (<47.00 and <48.00, respectively) group and high-PNI (≥47.00 and ≥48.00, respectively) group (Table 3). Also, we divided GPS values into two subgroups: GPS 0–1 as subgroup 1 and GPS 2 as subgroup 2.

There were no differences in age, gender, and area of residence (*p*-value ≥ 0.05). In CR-NET and CRC groups, we noticed that they had low PNI, more urban area patients, those in TNM stage III, and a GPS value of 2.

Regarding neurotransmitters, statistical differences (*p*-value < 0.05) were revealed between the two subgroups of PNI and GPS in DA and ST only in the CR-NET group: DA (*p*-value = 0.026 and *p*-value = 0.034, respectively) and ST values (*p*-value = 0.032 and *p*-value = 0.045, respectively) (Table 3).

In CR-NET and CRC groups, looking at the variation of the biological parameters, statistically significant variations were recorded among the NEU and ALB, between the two subgroups of PNI and GPS: CR-NET group NEU (PNI < 47.00 vs. PNI ≥ 47.00, *p*-value = 0.027; GPS1 vs. GPS2, *p*-value = 0.042) and ALB (PNI < 47.00 vs. PNI ≥ 47.00, *p* < 0.0001; GPS1 vs. GPS2, *p*-value < 0.0001); CRC group NEU (PNI < 48.00 vs. PNI ≥ 48.00, *p*-value = 0.041; GPS1 vs. GPS2, *p*-value = 0.048) and ALB (PNI < 48.00 vs. PNI ≥ 48.00, *p*-value <0.0001; GPS1 vs. GPS2, *p*-value < 0.0001).

Comparing the inflammatory indices in the CR-NET and CRC groups, in the two subgroups of PNI and GPS, statistically significant differences (*p*-value < 0.05) were revealed between the NLR (PNI < 47.00 vs. PNI ≥ 47.00, *p*-value = 0.044; GPS1 vs. GPS2, *p*-value = 0.008; GPS1 vs. GPS2, *p*-value = 0.041, in CRC group, respectively), PLR (PNI < 48.00 vs. PNI ≥ 48.00, *p*-value = 0.021; GPS1 vs. GPS2, *p*-value = 0.039, only in CRC group), and LMR (PNI < 47.00 vs. PNI ≥ 47.00, *p*-value = 0.054; GPS1 vs. GPS2, *p*-value = 0.058), reaching the significance limit.

### 2.4. Comparing Clinical Features of Different TNM Stages in CR-NET and CRC Groups

Table 4 displays how TNM stages and neurotransmitters, PNI, GPS, and SIR markers relate to each other.

In the CR-NET group, there were statistical differences in the majority of the indicators. Of them, the rise in TNM stages was associated with a decrease in Hb, ALB, and PNI but an increase in DA, ST, WBC, NEU, and NLR. In Figure 1, the associations between PNI and SIR markers at various TNM stages are displayed.

Using the one-way ANOVA test, we obtained that CRC patients had statistically significant higher values of NEU, NLR, PLR, and CRP, which increased in the three stages, while LMR, ALB, and PNI showed a decreasing trend.

### 2.5. Correlations between Neurotransmitters, PNI, and SIR Markers in CR-NET Group

Pearson’s correlation analysis revealed that the serum levels of DA correlated much better with PNI and SIR markers in the CR-NET group (Table 5).

Our study showed a statistically significant correlation between the DA levels and hematological index values, NLR and PLR (weak positive correlation, *r* = 0.302, *p*-value = 0.046, and weak negative correlation, *r* = −0.258, *p*-value = 0.038, respectively). Additionally, the correlation between the DA levels and PNI values reached the significance limit (weak negative correlation, *r* = −0.247, *p*-value = 0.053). The ST levels exhibited a moderate positive correlation with LMR values (*r* = 0.372, *p*-value = 0.047).

Another valuable point of our analysis was that both immunological and nutritional factors, PNI and GPS, correlated very strongly with ALB levels (negative correlations, *r* = −0.995, *p*-value < 0.0001, and *r* = −0.859, *p*-value = 0.047, respectively). Also, among these factors, only PNI was correlated moderately and negatively with NLR (*r* = −0.507, *p*-value = 0.009) and PLR (*r* = −0.438, *p*-value = 0.028).

We also registered statistically significant negative correlations between the serum levels of ALB and NLR (moderate correlation, *r* = −0.485, *p*-value = 0.009), PLR (moderate correlation, *r* = −0.438, *p*-value = 0.028), and LMR (moderate correlation, *r* = −0.466, *p*-value = 0.019).

## 3. Discussion

Our study analyzed for the first time the presence of DA, ST, NE, and EPI in the serum of patients with CR-NETs. We also determined the significance of PNI, GPS, and SIR markers as prognostic factors for patients with CR-NETs in different TNM stages. And last but not least, another novelty that our study brings is related to the identification of correlations between DA and PNI, as well as with NLR and PLR.

In order to identify and eradicate cancer cells, numerous studies have concentrated on the immune system and associated biological signaling pathways. It is believed that the brain–gut axis plays a significant role in the development and spread of GI malignancies [2]. The brain–gut axis denotes a brain-to-gut and gut-to-brain route that is bidirectional in its modulation of gastrointestinal function. The stomach can produce and secrete numerous neuroactive substances that can penetrate the blood–brain barrier and affect CNS activity [56]. The sympathetic and parasympathetic nervous systems, as well as the humoral pathway, can also transfer some neuroactive chemicals from the brain to the gut [57].

DA has an opposing effect on tumor growth despite being the precursor of EPI and NE. DA may have a beneficial or negative impact on carcinogenesis, in contrast to adrenaline and norepinephrine, which have been shown to promote tumor growth in a number of stress-induced malignancies [8].

According to Basu et al., CRC patients have lower levels of dopamine in the early stages of the disease than in normal tissues [58]. In another study, Chen et al. [59] demonstrated that dopamine enhanced the treatment efficacy of the traditional anti-tumor medication 5-FU in addition to having anti-tumor properties against CRC. Additionally, they thought that treating CRC with a combination of dopamine and 5-FU would offer a novel therapeutic approach [59].

In our study, we found that CR-NET patients showed significantly higher serum levels of DA compared to CRC patients (Table 2). We showed that serum DA was present in the early stages of CR-NETs, with increasing levels as we advanced through the TNM stages (Table 4, and Figure 1). Patients with increased expression of the dopamine receptor D2 (DRD2) had shorter survival times. Additionally, there was a negative correlation found between the expression of DRD2 and the prognosis of gastric cancer [60]. In gastric cancer tissue, DA levels were shown to be lower, and Chakroborty et al. discovered that DA supplementation inhibited angiogenesis, which in turn postponed the growth of gastric cancer [61]. Research indicates that people with cholangiocarcinoma have higher levels of DA secretion and that this rise is caused by the cancer cells themselves and promotes the growth of tumors. Cholangiocarcinoma cell proliferation can be inhibited in vitro by blocking DA production [62]. It was discovered that pancreatic cancer patients had higher levels of DRD2 expression and that by blocking the extracellular regulated kinase signaling pathway, DRD2 inhibitors could decrease the growth of tumors [63,64]. Patients with hepatocellular carcinoma (HCC) have lower serum levels of DA than do healthy, normal people. Nevertheless, administering DA to HCC cells can enhance their ability to proliferate, suggesting that DA plays a significant role in the formation of tumors [65].

We found a close relationship between the levels of DA and the inflammation and nutritional status of the CR-NET patients in this study, but we are unable to say that these findings are consistent with those of other studies due to the lack of pilot or prior data. Chemotherapy and radiation side effects often cause malnourishment in tumor patients, and it is also possible that tumor metastases into the gastrointestinal tract contribute to this problem. CRC patients are more likely to experience gastrointestinal metastases because they often occur in the digestive tract, which interferes with nutrition absorption and digestion [66]. CR-NET patients with moderate to severe malnutrition status (PNI < 47.00 subgroup) had a higher level of DA than those with mild malnutrition status (PNI ≥ 47.00 subgroup) (Table 3). Taking into account the fact that PNI is a composite score and the calculation formula based on a combination of serum ALB level and the total number of LYM, we noticed that patients from the PNI < 47.00 subgroup who had lower serum ALB levels, corresponding to nutritional status deficiency, had an increased DA level. Also, patients who had a serum CRP level above 10 mg/dL and ALB below 3.5 mg/dL, corresponding to an increased inflammatory status and a progressive nutritional decline, were included in the hypoalbuminemia. This means that DA can represent a diagnostic and a prognostic marker. The correlations between DA levels and hematological indices obtained also support these findings from our study. Our investigation revealed a weak but statistically significant positive correlation between serum DA levels and NLR and PLR (Table 5).

The development of a systemic inflammatory response in cancer patients is characterized by elevated CRP and hypoalbuminemia, which has a substantial impact on survival due to accelerated protein breakdown and direct catabolic effects on skeletal muscles and other host tissues. In many cancers, a systemic inflammatory response is considered to be predictive of tumor invasion, metastasis, and angiogenesis, all of which are linked with a poor prognosis [28,29,30,31,32,33,34,35,36,37,38,39,40,41].

One of the predominant leukocyte subsets in peripheral blood, neutrophils (NEU) actively contribute to the development, spread, and metastasis of tumors [21]. According to Zhang et al., by inhibiting peripheral leukocyte activation, circulating tumor-associated NEU can increase the survival time of tumor cells [67]. Conversely, a prolonged and high NEU count following surgery is thought to be associated with the occurrence of recurrence and to foster the establishment of micrometastatic lesions [28]. NEU may act as mediators of lymph node metastasis in Pan-NETs, according to Tong et al. [21]. On the other hand, lymphocytes (LYM) are a component of the host immune system and are crucial to anti-tumor immunity [68]. It is thought that the whole immune system reactivity of a cancer patient is reflected in the absolute LYM count. Thus, a reduction in lymphocytes is thought to be associated with recurrence and decreased survival [69].

Our study showed that CR-NET patients had significantly higher values of WBC, NEU, and LYM, compared to CRC patients (Table 2). We showed that NEU values were present in the early stages of CR-NETs, with increasing levels as we advanced through the TNM stages (Table 4). In this study, CR-NET patients with hypoalbuminemia (PNI < 47.00 subgroup) and inflammatory status (GPS 2) had a higher value of NEU than those with PNI ≥ 47.00 and GPS 0–1 (Table 3).

The previously indicated impacts of LYMs and NEUs may work in concert with the NLR to make it a potentially useful biomarker for tumor prognosis. Our research showed that NLR can be an independent prognostic factor for CR-NET patients with hypoalbuminemia (PNI < 47.00 sub-group), inflammatory status (GPS 2), and TNM stage III.

Recent research has examined the relationship between poor prognosis in various oncological tumors and the NLR and PLR [35,42,43,44,45,46,47].

Since cancer patients can easily be tested for NLR and PLR in peripheral blood, researchers have investigated their ability to predict prognostic outcomes and risk classification prior to therapy. Researchers have used NLR and PLR in numerous investigations on the prognosis of NETs. Research demonstrating the connection between PLR, NETs, and NLR typically relates to survival and clinical characteristics. It was found that in NETs, a high NLR was associated with larger tumor sizes, advanced stages, high grades, and shorter survival times [45]. Preoperative NLR was identified as a possible independent indicator of lymph node metastasis and recurrence-free survival in another study [21]. Some studies have shown no correlation between the increase in PLR and metastasis, despite others showing an increase in NLR and PLR as the grade increases in NETs [35,42,47]. A threshold NLR of 2.6 predicted the presence of peritoneal metastasis with a specificity of 100%, sensitivity of 71%, and overall accuracy of 84%. NLR was considerably greater in patients with distant metastasis [42].

A few studies comparing NLR and PLR in NETs using just histopathological parameters were found during a literature search. The association between NLR and the Ki-67 proliferation index, lymphovascular invasion, and histological grade in NETs was found in a number of investigations [46,70]. According to a study by Kulahci et al., patients with NETs who had NLR cutoff values greater than 3.01 had higher histological grades, higher Ki-67 proliferation indices, higher mitosis, and more lymphovascular invasion [44].

In our study, Pearson’s correlation analysis revealed that the NLR and PLR values were correlated with PNI values, moderately and negatively. Moreover, both hematological indices were negatively correlated with ALB (Table 5). In the CR-NET group, the cutoff values were determined to be the median NLR, PLR, and LMR, which were 2.58, 118.67, and 4.31, respectively. We noticed that in the PNI < 47.00 subgroup, more patients with CR-NET had NLR over 2.58, PLR under 118.67, and hypoalbuminemia (Table 3). Another valuable point of our Pearson’s analysis was that other immunological and nutritional factors that we studied, such as GPS, correlated very strongly with ALB levels. Also, we observed that in the GPS 2 subgroup, more patients with CR-NET had NLR over 2.58, LMR over 4.31, and hypoalbuminemia (Table 3). This means that PNI can serve as a diagnostic and prognostic marker by reflecting both the nutritional and immune status of CR-NET patients.

There is a lack of research suggesting that prognostic indicators based on inflammation, such as PNI and GPS, provide survival information for patients with CR-NETs. To our knowledge, no study has examined the integration of PNI, SIR markers, and GPS as an additional index, based on the existing TNM staging system in CR-NETs.

Few studies have investigated the role of PNI and GPS in patients with CRC [20,24,26,27]. Nakamoto et al. conducted a study to investigate the prognostic value of seven inflammation-based biomarkers—NLR, PLR, LMR, SIRI, GPS, PNI, systemic immune-inflammation index (SII), and systemic inflammation response index (SIRI)—for recurrence following curative surgery in patients with CRC and to further elucidate the clinical significance of clinicopathological factors that include these biomarkers [26]. Only Bai et al., in a study published in 2020, investigated the relationship between PNI, SIR markers, and GPS and TNM staging and metastasis of CRC. They reported that we can diagnose clinical stage, metastasis, and prognostic cofactors for evaluation if we combine the detection of PNI, GPS, and SIR levels and correlate them with TNM staging and lymph node metastasis [20]. Regardless of whether patients have received radical or palliative treatment, a meta-analysis by He et al. found that the GPS or modified Glasgow Prognostic Score (mGPS), which is derived from two common laboratory serum indicators, is an independent and promising indicator for predicting the prognostic outcomes of CRC patients [24].

The findings of our investigation, which were previously discussed, relating to the PNI, GPS, SIR, and DA under investigation indicate that these tools provide comprehensive markers of nutritional and systemic inflammation status, are simple to use, and are repeatable. Clinically, the PNI, GPS, SIR, and DA can be easily and affordably collected from patients, and they may be used as biomarkers in the management of CR-NET patients. Further research on this topic could provide valuable insights into which biomarkers to incorporate into clinical practice for the management of CR-NET patients.

Since this study was restricted to our reference center, we acknowledge that it has inherent limitations. The time constraint imposed on the study: we chose to conduct a prospective study over a period of two years, and in this study period, only 25 patients were diagnosed with CR-NETs. We achieved statistical significance even with 25 patients in the study group; therefore, we do not think it is necessary to debate the simulation of sample size determination at this time. In addition, as we lacked a pilot study and prior data that were referenced in the literature, we were unable to determine the effect size and hence did not compute the sample size. The limit on resources—the requirement to finish the PhD thesis—was another limiting factor.

## 4. Materials and Methods

This study included 25 consecutive patients who were pathologically diagnosed with CR-NETs and an initial control group consisting of 125 patients with newly diagnosed CRC that met the inclusion criteria, with similar ages, urban/rural areas, and female/male ratios. The study was conducted in accordance with the Declaration of Helsinki and approved by the Ethics Committee of the University of Medicine and Pharmacy of Craiova (no. 63/28 April 2021).

The diagnosis and reporting of CR-NET and CRC patients were carried out according to the latest criteria developed by the World Health Organization (WHO) working group for tumors of the digestive system [71].

### 4.1. Patient Selection

Inclusion criteria: patients aged 18 years or older, who had a confirmed diagnosis of CR-NETs or CRC, supported by a histopathological (HP) result containing information on tumor type, grading, and classification in the pTNM system; those who have not had radiation therapy, chemotherapy, biological therapies, or immunotherapy; those free of autoimmune illnesses; and those without significant liver or renal diseases. Informed consent provided by the patients was also a requirement.

The patients were diagnosed at the Emergency County Clinical Hospital of Craiova.

Medical documentation was used to gather and analyze demographic and clinical data about patients. The initial evaluation for every patient comprised the following details: contact details, age, gender, place of residence, date of diagnosis, tumor localization, microscopic findings, pathological staging, lymph nodes and distant metastases, kind of complication, and available treatment options.

We analyzed the biological constants (complete blood count, biochemistry, and coagulation tests) paraclinically. The following procedures were recorded in order to get the most thorough evaluation: plain abdominal X-ray, chest X-ray, abdominal-pelvic ultrasonography, thoracic-abdominal-pelvic computer-tomographic examination with contrast material given intravenously and orally, and, when necessary, magnetic resonance imaging examination, colonoscopy, and digestive endoscopy.

We also considered the characteristics and clinical data of other patients, including their environment (urban or rural), lifestyle (obesity, smoking, heavy alcohol consumption), educational level (primary school, high school, higher education), and family history (colorectal cancer, other digestive cancers, other types of cancers).

After the HP examination, we noticed that in the CR-NET group patients, we did not identify poorly differentiated (G3 NET) tumors. Additionally, we found no metastases in any patient, indicating the absence of TNM stage IV cases. Based on the obtained results, we excluded 43 patients with poorly differentiated (G3) tumors and 22 patients with TNM stage IV diagnoses from the CRC group. Finally, we limited the number of CRC patients in the statistical analysis to 60.

### 4.2. Sample Collection

Approximately 5 mL of venous blood was drawn from patients in both groups and placed into tubes without any additional ingredients (Vacutest Kima, Arzegrande, Padova, Italy). The normal protocol called for centrifugation (3.000× *g* for 10 min) to separate the clot as soon as possible after harvesting but no later than 4 h. To process the samples over a longer period of time, we coded the serum sample tubes for each patient, sealed them to prevent contamination, and stored them at temperatures between −20 °C and even −80 °C. Freezing–unfreezing cycle activities were avoided, and the frozen samples were allowed to come to room temperature before working with the patient samples.

### 4.3. Calculation of Systemic Inflammatory Response Markers

An automatic hematology analyzer was used to separate and count all five types of white blood cells (neutrophils, monocytes, lymphocytes, basophils, and platelets). Using flow cytometry and Coulter’s principle, we were able to obtain an extended leukocyte formula of 5 diff (Ruby Cell-Dyne, Abbott, Abbott Park, IL, USA). We computed the systemic inflammatory response (SIR) markers derived from the blood cell count, NLR, MLR, and PLR using these determinations.

We conducted ESR using the Westergren method (ESR tubes, Becton Dickinson, USA). We determined CRP and ALB using the chemiluminescence immunological technique and an automatic immunoassay analyzer (Cobas e411, Roche Diagnostics GmbH, Mannheim, Germany).

### 4.4. Immunological Assessment

At the Immunology Laboratory at the University of Medicine and Pharmacy of Craiova, the Enzyme-Linked Immunosorbent Assay (ELISA) technique was utilized to quantitatively determine the serum levels of DA, ST, NE, and EPI.

For each of the following mediators, we used test sets that were accessible commercially: DA, EPI (Catalog No: E-EL-0046, E-EL-0045; Sensitivity: 18.75 pg/mL; Detection Range: 31.25–2000 pg/mL), NE (Catalog No:E-EL-H0047; Sensitivity: 0.19 ng/mL; Detection Range: 0.31–20 ng/mL), and ST (Catalog No: E-EL-0033; Sensitivity: 9.38 ng/mL; Detection Range: 15.63–1000 ng/mL), Elabscience (Houston, TX, USA).

We followed the manufacturer’s instructions and suggested procedures, dilutions, and operating procedures. We used the ELISA method with a standard optical analyzer (Asys Expert Plus UV G020 150 Microplate Reader, ASYS Hitech GmbH, Eugendorf, Austria) with a 450 nm wavelength.

### 4.5. Calculation of Prognostic Nutritional Index and Glasgow Prognostic Score

The Prognostic Nutritional Index (PNI) is based on serum ALB level and absolute lymphocyte count. The PNI was calculated according to the acknowledged formula: 10 × serum albumin (g/dL) + 0.5% × total lymphocyte number (per mm^3^) [72,73]. Interpretation: PNI value ≥ 50, normal; PNI value < 50, mild malnutrition; PNI value < 45, moderate to severe malnutrition; PNI value < 40, severe malnutrition.

Glasgow Prognostic Score (GPS) was based on CRP and ALB levels, patients with CRP ≤ 10 mg/L and ALB ≥ 35 g/L were allocated to GPS-0 group. Patients with only CRP > 10 mg/L were assigned to GPS-1 group. Patients who had both CRP > 10 mg/L and ALB < 35 g/L were allocated to GPS-2 group [74].

The median NLR, PLR, and LMR were 2.58, 118.67, and 4.31, respectively, and were chosen as the cutoff in the CR-NET group. In the CRC group, we determined the following medians: 2.95, 148.38, and 3.48, respectively. Provided that 47.00 was the median value among the 25 CR-NET patients and 48.00 for the 60 CRC patients, we used the median of PNI scores as classified criteria that were divided into two groups: low-PNI (<47.00 and <48.00, respectively) group and high-PNI (≥47.00 and ≥48.00, respectively) group.

### 4.6. Statistical Analysis

We managed and processed patient data from medical documents using Microsoft Excel 2021. We used GraphPad Prism 5 Version (LLC, San Diego, CA, USA) to analyze the data.

The data were checked for normality using the D’Agostino and Pearson omnibus normality tests.

The means of the following variables are displayed together with the standard deviation (SD): DA, ST, NE, EPI, NLR, PLR, and LMR all had normal distributions. CRP, ALB, and PNI were shown to have non-normal distributions; the results are shown as the median with inter-quartile range. We expressed the category values as percentages.

Using the Mann–Whitney U test or the Kruskal–Wallis H test (used for non-Gaussian distributions), continuous variables were analyzed to determine the difference between groups.

Pearson’s coefficients (−1 < *r* < 1) were used to see if there were any significant correlations between the levels of DA, ST, NE, EPI, NLR, PLR, LMR, CRP, ALB, GPS, and PNI.

## 5. Conclusions

In summary, this study revealed that CR-NET patients showed significantly higher serum levels of DA compared to CRC patients. We showed that serum DA was present in the early stages of CR-NETs, with increasing levels as we advanced through the TNM stages. Moreover, we found a close relationship between the levels of DA and the inflammation and nutritional status of the CR-NET patients in this study. The findings of our investigation, which were previously discussed, relating to the PNI, GPS, SIR, and DA under investigation indicate that these tools provide comprehensive markers of nutritional and systemic inflammation status, are simple to use, and are repeatable. Clinically, the PNI, GPS, SIR, and DA can be easily and affordably collected from patients, and they may be used as biomarkers in the management of CR-NET patients. Our findings can certainly constitute a starting point for future research and extended studies with multicentric involvement.

## Figures and Tables

**Figure 1 ijms-25-06977-f001:**
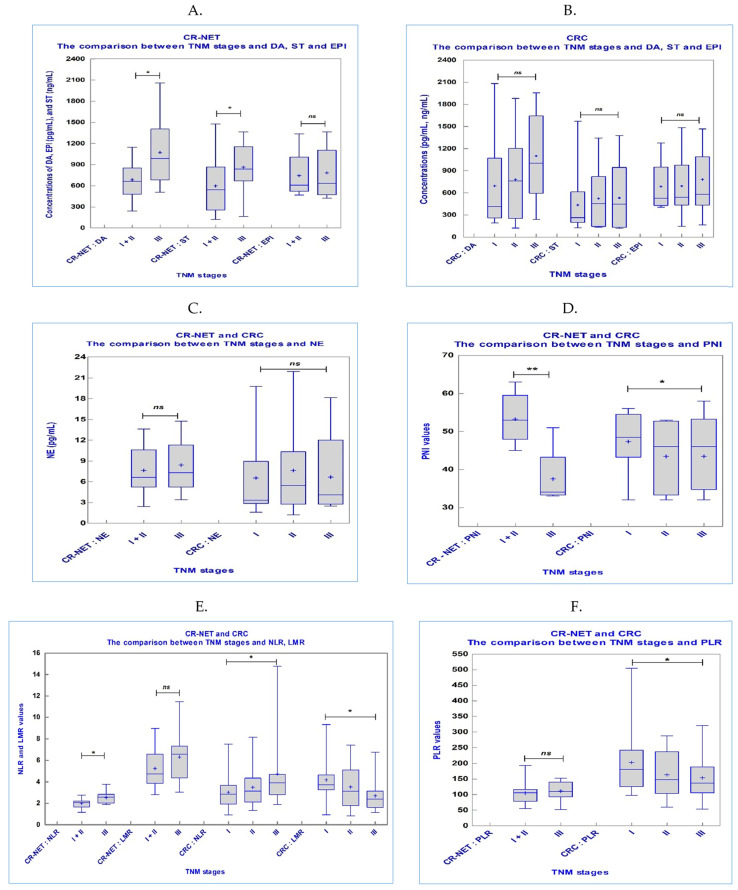
Comparison between TNM stage and biomarkers in study groups: (**A**) serum levels of DA (pg/mL), ST (ng/mL), and EPI (pg/mL) in CR-NET group; (**B**) serum levels of DA (pg/mL), ST (ng/mL), and EPI (pg/mL) in CRC group; (**C**) serum levels of NE (pg/mL) in CR-NET and CRC groups; (**D**) PNI values in CR-NET and CRC groups; (**E**) NLR and LMR values in CR-NET and CRC groups; and (**F**) PLR values in CR-NET and CRC groups. Plot bars represent serum levels/values of parameters from individual samples; horizontal lines represent mean values accompanied by the standard deviation (SD) represented by whiskers; * *p* < 0.05; ** *p* < 0.0001; ns: not statistically significant; I: TNM stage I; II: TNM stage II; III: TNM stage III.

**Table 1 ijms-25-06977-t001:** Patient demographic and clinical characteristics.

Characteristics	CR-NET Group (*n* = 25)	CRC Group (*n* = 60)
Age (yrs) (mean ± SD)	64.52 ± 11.13	68.75 ± 8.46
Gender, Female/Male (*n*)	11/14	29/31
Area of residence, Rural/Urban (*n*)	10/15	31/29
Tumor extension (pT) (*n*)		
T1	7	12
T2	9	18
T3	6	20
T4	3	10
Regional lymph node metastasis (pN) (*n*)		
N0	13	38
N1	12	22
Distant metastasis (pM) (*n*)		
M0	25	60
M1	0	0
TNM Stage of WHO Classification of Tumors 2019 (*n*)		
I	6	12
II	7	14
III	12	34
Tumor grade (G) (*n*)	WHO classification of GI NETs 2017	WHO classification of Tumors 2019
	G1 NET—14	G1—34
	G2 NET—11	G2—26
Locations of tumor (*n*)		
Appendix	5	-
Cecum	-	6
Ascending colon	4	9
Hepatic flexure	4	6
Transverse colon	3	3
Splenic flexure	2	8
Descending colon	-	4
Sigmoid colon	5	19
Rectosigmoid junction	2	3
Rectum	-	2
Complications (*n*)		
Hemorrhage	2	7
Obstruction	4	12
Perforation	2	8

CRC: colorectal cancer; CR-NET: colorectal neuroendocrine tumor; GI: gastrointestinal; TNM: tumor–node–metastasis; WHO: World Health Organization; G1: well differentiated; G2: moderately differentiated.

**Table 2 ijms-25-06977-t002:** Mean, median, and statistical significance of neurotransmitters, PNI, GPS, and SIR markers in CR-NET and CRC groups.

Parameter (Mean ± SD)	CR-NET Group (*n* = 25)	CRC Group (*n* = 60)	*p*-Value from Student’s *t*-Test
DA (pg/mL)	871.53 ± 411.93	796.09 ± 588.48	0.043 *
NE (pg/mL)	8.02 ± 3.54	7.19 ± 5.92	0.138
EPI (pg/mL)	765.48 ± 308.38	733.71 ± 366.64	0.246
ST (ng/mL)	724.62 ± 396.01	477.36 ± 358.60	0.021 *
Hb (g/dL)	10.64 ± 2.38	10.49 ± 2.09	0.875
WBC (×10^3^/μL)	9.91 ± 2.24	8.60 ± 1.96	0.046 *
NEU (×10^3^/μL)	6.30 ± 1.84	5.77 ± 1.74	0.048 *
LYM (×10^3^/μL)	2.82 ± 0.67	1.98 ± 0.79	0.039 *
MON (×10^3^/μL)	0.54 ± 0.22	0.61 ± 0.23	0.124
PLT (×10^3^/μL)	291.52 ± 80.98	284.6 ± 78.06	0.334
NLR	3.22 ± 1.82	3.44 ± 2.18	0.204
PLR	136.77 ± 59.76	165.80 ± 82.24	0.031 *
LMR	4.97 ± 2.28	3.73 ± 1.86	0.045 *
ESR (mm/1st h)	58.48 ± 27.64	45.17 ± 31.42	0.003 *
GPS (*n*)			
0–1	16	39	-
2	9	21	-
Parameter [median (range)]			
CRP (mg/dL)	40.00 (20.00–81.00)	18.50 (7.50–61.20)	0.027 *
ALB (g/dL)	4.70 (3.30–6.30)	4.80 (3.20–5.80)	0.392
PNI	47.00 (33.01–63.02)	48.00 (32.01–58.02)	0.511

CRC: colorectal cancer; CR-NET: colorectal neuroendocrine tumor; ALB: albumin; DA: dopamine; ST: serotonin; EPI: adrenaline/epinephrine; NE: noradrenaline/norepinephrine; Hb: hemoglobin; CRP: C-reactive protein; ESR: erythrocyte sedimentation rate; WBC: white blood cells/leukocytes; NEU: neutrophils; LYM: lymphocytes; MON: monocytes; PLT: platelets; NLR: neutrophil/lymphocyte ratio; LMR: lymphocyte/monocyte ratio; PLR: platelet/lymphocyte ratio; PNI: Prognostic Nutritional Index; GPS: Glasgow Prognostic Score; SIR: systemic inflammatory response; SD: standard deviation; * *p* < 0.05: statistically significant.

**Table 3 ijms-25-06977-t003:** Comparing the PNI and GPS groups’ clinical features between the study groups.

**Variables**	**CR-NET Group**								**CRC Group**						
	**All** **Patients**	PNI			GPS				**All** **Patients**	PNI			GPS		
		PNI < 47	PNI ≥ 47	*p*	1	2	*p*			PNI < 48	PNI ≥ 48	*p*	1	2	*p*
Patients (*n*)	25	12	13		16	9			60	34	26		39	21	
Age (yrs)	64.52 ± 11.13	61.83 ± 10.96	67.00 ± 11.33	0.271	66.06 ± 11.98	61.78 ± 9.44	0.136	Age (yrs)	68.75 ± 8.46	68.32 ± 8.74	69.50 ± 5.22	0.671	69.15 ± 8.15	68.00 ± 8.78	0.572
<66	12	6	6		7	5		<69	19	16	3		16	11	
≥66	13	6	7		9	4		≥69	41	18	23		23	10	
Gender, Female/Male (*n*)	11/14	4/8	7/6		9/7	2/7			29/31	19/15	10/16		18/21	11/10	
Area of residence, Rural/Urban (*n*)	10/15	3/8	7/7		8/8	2/7			31/29	15/19	16/10		22/17	9/12	
TNM stages (*n*)															
I	6	1	5		6	-			12	7	5		7	5	
II	7	1	6		7	-			14	7	7		12	2	
III	12	10	2		3	9			34	20	14		20	14	
GPS (*n*)															
0–1	16	3	13		-	-			39	13	26		-	-	
2	9	9	0		-	-			21	21	0		-	-	
NLR (mean ± SD), (*n*)	3.22 ± 1.82	2.83 ± 0.48	1.95 ± 0.68	0.044 *	1.95 ± 0.38	2.47 ± 0.61	0.008 *		3.44 ± 2.18	3.18 ± 1.43	3.78 ± 2.88	0.538	2.75 ± 1.11	3.81 ± 2.51	0.041 *
<2.58	15	5	10		13	2		<2.95	28	20	8		23	5	
≥2.58	10	7	3		3	7		≥2.95	32	14	18		16	16	
PLR (mean ± SD), (*n*)	136.77 ± 59.76	101.51 ± 29.17	113.71 ± 37.18	0.225	110.10 ± 36.28	102.09 ± 28.86	0.729		165.80 ± 82.24	132.48 ± 75.72	176.12 ± 91.49	0.021 *	168.50 ± 56.87	135.45 ± 74.45	0.039 *
<118.67	16	8	8		10	6		<148.38	31	23	8		19	12	
≥118.67	9	4	5		6	3		≥148.38	29	11	18		20	9	
LMR (mean ± SD), (*n*)	4.97 ± 2.28	5.34 ± 1.78	4.62 ± 2.28	0.054 *	5.48 ± 1.67	6.23 ± 2.53	0.058 *		3.73 ± 1.86	3.04 ± 1.87	3.11 ± 1.89	0.984	3.04 ± 1.58	3.19 ± 1.15	0.729
<4.31	8	4	4		5	3		<3.48	30	15	15		24	6	
≥4.31	17	8	9		11	6		≥3.48	30	19	11		15	15	
DA (pg/mL) (mean ± SD), (*n*)	871.53 ± 411.93	972.36 ± 311.02	770.62 ± 352.34	0.026 *	822.96 ± 459.32	920.60 ± 299.93	0.034 *		796.09 ± 588.48	812.90 ± 471.57	784.80 ± 555.40	0.452	738.10 ± 577.20	854.20 ± 512.22	0.741
<814.13	12	5	7		9	3		<618.63	30	17	13		22	8	
≥814.13	13	7	6		7	6		≥618.63	30	17	13		17	13	
NE (pg/mL) (mean ± SD), (*n*)	8.02 ± 3.54	8.32 ± 3.23	7.72 ± 3.63	0.102	7.62 ± 3.48	8.43 ± 3.09	0.094		7.19 ± 5.92	8.13 ± 4.14	6.26 ± 5.85	0.068	6.91 ± 5.77	6.48 ± 5.56	0.725
<7.22	12	6	6		7	5		<4.64	26	15	11		19	7	
≥7.22	13	6	7		9	4		≥4.64	34	19	15		20	14	
EPI (pg/mL) (mean ± SD), (*n*)	765.48 ± 308.38	780.78 ± 278.85	751.36 ± 344.19	0.635	756.98 ± 322.53	780.57 ± 279.76	0.647		733.71 ± 366.64	725.70 ± 376.00	742.30 ± 362.80	0.594	747.30 ± 350.60	717.10 ± 391.50	0.569
<632.56	12	5	7		8	4		<545.39	30	18	12		20	10	
≥632.56	13	7	6		8	5		≥545.39	30	16	14		19	11	
ST (ng/mL) (mean ± SD), (*n*)	724.62 ± 396.01	829.77 ± 211.56	618.33 ± 314.35	0.032 *	606.11 ± 408.53	842.60 ± 504.20	0.045 *		477.36 ± 358.60	518.97 ± 307.10	439.40 ± 334.30	0.052	458.76 ± 349.40	500.64 ± 332.33	0.101
<713.38	12	5	7		9	3		<403.58	28	16	12		18	10	
≥713.38	13	7	6		6	7		≥403.58	32	18	14		21	11	
Hb (g/dL) (mean ± SD)	10.64 ± 2.38	10.41 ± 2.23	10.86 ± 2.58	0.611	10.95 ± 2.17	10.10 ± 1.72	0.972		10.49 ± 2.09	10.56 ± 2.00	10.42 ± 2.26	0.777	10.32 ± 2.07	10.83 ± 2.15	0.382
WBC (×10^3^/μL) (mean ± SD)	9.91 ± 2.24	10.33 ± 2.01	9.53 ± 2.45	0.656	9.75 ± 1.85	10.00 ± 2.48	0.318		8.60 ± 1.96	8.44 ± 2.03	8.81 ± 1.88	0.372	7.98 ± 2.09	8.93 ± 1.83	0.238
NEU (×10^3^/μL) (mean ± SD)	6.30 ± 1.84	7.08 ± 1.69	5.53 ± 1.76	0.027 *	5.77 ± 1.32	6.60 ± 2.06	0.042 *		5.77 ± 1.74	7.36 ± 1.58	5.89 ± 2.04	0.041 *	5.91 ± 1.95	7.55 ± 10.26	0.048 *
LYM (×10^3^/μL) (mean ± SD)	2.82 ± 0.67	3.05 ± 0.49	2.62 ± 0.76	0.158	2.72 ± 0.73	3.00 ± 0.54	0.222		2.00 ± 0.79	2.10 ± 1.82	1.65 ± 0.76	0.180	1.67 ± 0.72	2.13 ± 1.95	0.220
MON (×10^3^/μL) (mean ± SD)	0.54 ± 0.22	0.58 ± 0.24	0.51 ± 0.21	0.534	0.54 ± 0.20	0.56 ± 0.26	0.826		0.61 ± 0.35	0.69 ± 0.43	0.53 ± 0.20	0.105	0.55 ± 0.21	0.69 ± 0.46	0.230
PLT (×10^3^/μL) (mean ± SD)	291.52 ± 81.00	281.87 ± 83.47	301.98 ± 80.51	0.596	294.00 ± 48.38	290.14 ± 96.13	0.996		284.6 ± 78.06	278.20 ± 81.72	290.60 ± 75.33	0.600	281.40 ± 78.40	288.50 ± 78.90	0.846
CRP (mg/dL) [median (range)]	40.00 (20.00–81.00)	36.45 (24.00–81.00)	43.57 (20.00–78.50)	0.056 *	39.65 (20.00–78.50)	40.37 (24.00–81.00)	0.337		18.50 (7.50–61.20)	19.00 (9.31–57.60)	17.95 (7.50–61.20)	0.689	20.00 (7.50–61.20)	16.30 (9.31–48.00)	0.101
ALB (g/dL) [median (range)]	4.70 (3.30–6.30)	3.40 (3.30–4.60)	5.30 (4.70–6.30)	<0.0001 *	5.10 (4.50–6.30)	3.40 (3.30–3.50)	<0.0001 *		4.48 (3.20–5.80)	3.50 (3.20–4.80)	5.30 (4.80–5.80)	<0.0001 *	5.10 (3.50–5.80)	3.40 (3.25–3.50)	<0.0001 *

ALB: albumin; DA: dopamine; ST: serotonin; EPI: adrenaline/epinephrine; NE: noradrenaline/norepinephrine; Hb: hemoglobin; CRP: C-reactive protein; WBC: white blood cells/leukocytes; NEU: neutrophils; LYM: lymphocytes; MON: monocytes; PLT: platelets; NLR: neutrophil/lymphocyte ratio; LMR: lymphocyte/monocyte ratio; PLR: platelet/lymphocyte ratio; PNI: Prognostic Nutritional Index; GPS: Glasgow Prognostic Score; SD: standard deviation; * statistically significant.

**Table 4 ijms-25-06977-t004:** The comparison between TNM stage and biomarkers in study groups.

Variables (Mean ± SD)	CR-NET			CRC			
	TNM Stage I + II (*n* = 13)	TNM Stage III (*n* = 12)	*p*-Value from Student’s *t*-Test	TNM Stage I (*n* = 12)	TNM Stage II (*n* = 14)	TNM Stage III (*n* = 34)	*p*-Value from One-Way ANOVA
DA (pg/mL)	687.07 ± 270.39	1071.37 ± 454.62	0.024 *	694.7 ± 583.7	782.4 ± 578.6	1099.0 ± 554.2	0.121
NE (pg/mL)	7.67 ± 3.42	8.40 ± 3.78	0.527	6.55 ± 5.86	6.69 ± 5.71	7.63 ± 6.15	0.808
EPI (pg/mL)	744.49 ± 284.60	784.85 ± 339.27	0.558	686.1 ± 308.4	692.1 ± 373.1	768.0 ± 392.0	0.716
ST (ng/mL)	596.70 ± 418.60	863.17 ± 332.77	0.008 *	254.3 ± 416.2	436.3 ± 362.1	531.3 ± 433.1	0.669
Hb (g/dL)	11.40 ± 2.88	9.48 ± 1.56	0.038 *	10.79 ± 2.18	10.40 ± 2.27	10.44 ± 2.05	0.869
WBC (×10^3^/μL)	8.19 ± 1.44	10.24 ± 2.67	0.046 *	8.48 ± 1.94	8.56 ± 2.07	8.76 ± 1.82	0.937
NEU (×10^3^/μL)	5.33 ± 1.32	7.50 ± 2.37	0.034 *	5.64 ± 1.85	5.79 ± 1.34	6.11 ± 1.92	0.050 *
LYM (×10^3^/μL)	2.71 ± 0.75	2.95 ± 0.57	0.098	2.15 ± 0.75	2.01 ± 0.92	1.58 ± 0.63	0.097
MON (×10^3^/μL)	0.56 ± 0.21	0.53 ± 0.24	0.873	0.65 ± 0.26	0.58 ± 0.23	0.64 ± 0.19	0.549
PLT (×10^3^/μL)	268.50 ± 82.99	316.40 ± 74.13	0.293	294.40 ± 83.11	270.70 ± 84.36	273.20 ± 53.55	0.548
ESR (mm/1st h)	44.84 ± 20.98	71.08 ± 27.69	0.049 *	29.50 ± 14.37	47.29 ± 36.93	47.41 ± 25.65	0.132
NLR	1.98 ± 0.43	2.56 ± 0.59	0.030 *	2.97 ± 1.53	3.48 ± 1.75	4.74 ± 3.50	0.043 *
PLR	104.93 ± 37.15	111.03 ± 30.28	0.107	153.50 ± 69.60	163.80 ± 80.55	203.10 ± 109.90	0.039 *
LMR	5.25 ± 1.75	6.32 ± 2.19	0.134	4.17 ± 1.79	3.52 ± 2.02	2.70 ± 1.52	0.045 *
**Variables** [median (range)]							
CRP (mg/dL)	43.00 (28.50–78.50)	32.45 (20.00–81.00)	0.904	16.15 (7.50–48.00)	17.31 (12.00–36.00)	31.25 (15.80–61.20)	<0.0001 *
ALB (g/dL)	5.30 (4.70–6.30)	3.40 (3.30–5.10)	<0.0001 *	4.85 (3.20–5.60)	4.65 (3.20–5.30)	4.25 (3.20–5.33)	0.038 *
PNI	53.01 (45.01–6.30)	34.01 (33.01–51.01)	<0.0001 *	48.50 (32.01–56.02)	46.50 (32.01–53.02)	42.50 (32.01–53.32)	0.038 *

ALB: albumin; DA: dopamine; ST: serotonin; EPI: adrenaline/epinephrine; NE: noradrenaline/norepinephrine; Hb: hemoglobin; CRP: C-reactive protein; ESR: erythrocyte sedimentation rate; WBC: white blood cells; NEU: neutrophils; LYM: lymphocytes; MON: monocytes; PLT: platelets; NLR: neutrophil/lymphocyte ratio; LMR: lymphocyte/monocyte ratio; PLR: platelet/lymphocyte ratio; PNI: Prognostic Nutritional Index; SD: standard deviation; * *p* < 0.05: statistically significant.

**Table 5 ijms-25-06977-t005:** Correlations between neurotransmitters, PNI, and SIR markers in CR-NET group.

	NE	EPI	ST	ALB	NLR	PLR	LMR	PNI	GPS
DA	*r* = 0.241 *p* = 0.045 *	*r* = 0.189 *p* = 0.041 *	*r* = 0.076 *p* = 0.718	*r* = −0.247 *p* = 0.132	*r = 0.302* *p = 0.046* *	*r = −0.258* *p = 0.038* *	*r* = 0.228 *p* = 0.273	*r = −0.247* *p = 0.053 ***	*r* = 0.074 *p* = 0.723
NE		*r* = 0.244 *p* = 0.240	*r* = 0.200 *p* = 0.336	*r* = −0.175 *p* = 0.144	*r* = 0.135 *p* = 0.517	*r* = −0.065 *p* = 0.756	*r* = 0.410 *p* = 0.141	*r* = 0.174 *p* = 0.403	*r* = 0.269 *p* = 0.194
EPI			*r* = −0.136 *p* = 0.517	*r* = −0.110 *p* = 0.601	*r* = 0.072 *p* = 0.733	*r = 0.066* *p = 0.751*	*r* = −0.147 *p* = 0.484	*r* = 0.109 *p* = 0.601	*r* = 0.018 *p* = 0.933
ST				*r* = −0.325 *p* = 0.155	*r* = 0.210 *p* = 0.338	*r = −0.216* *p = 0.299*	*r* = 0.372 *p* = 0.047 *	*r* = 0.125 *p* = 0.549	*r* = 0.447 *p* = 0.225
ALB					*r = −0.485* *p = 0.009* *	*r* = −0.438 *p* = 0.028 *	*r* = −0.466 *p* = 0.019 *	*r* = −0.995 *p* < 0.0001 *	*r* = −0.859 *p* = 0.047 *
NLR						*r* = 0.654 *p* < 0.0001 *	*r* = −0.547 *p* = 0.124	*r* = −0.507 *p* = 0.009 *	*r* = 0.292 *p* = 0.155
PLR							*r* = −0.597 *p* = 0.002 *	*r* = −0.438 *p* = 0.028 *	*r* = 0.105 *p* = 0.616
LMR								*r* = 0.466 *p* = 0.118	*r* = 0.560 *p* = 0.978
PNI									*r* = 0.159 *p* = 0.447

ALB: albumin; DA: dopamine; ST: serotonin; EPI: adrenaline/epinephrine; NE: noradrenaline/norepinephrine; NLR: neutrophil/lymphocyte ratio; LMR: lymphocyte/monocyte ratio; PLR: platelet/lymphocyte ratio; PNI: Prognostic Nutritional Index; GPS: Glasgow Prognostic Score; * *p* < 0.05: statistically significant; ** reached the significance limit.

## Data Availability

The data used to support the findings of this study are available from the corresponding author upon reasonable request.

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
