# Peer review of "Correlation between Neurotransmitters (Dopamine, Epinephrine, Norepinephrine, Serotonin), Prognostic Nutritional Index, Glasgow Prognostic Score, Systemic Inflammatory Response Markers, and TNM Staging in a Cohort of Colorectal Neuroendocrine Tumor Patients"

_ijms, 2024, doi:10.3390/ijms25136977_

Round 1
Reviewer 1 Report
Comments and Suggestions for Authors
Reviewer suggestions and comments
The authors in this study examined the presence and clinical application of serum DA, ST, NE, and EPI, in addition to determining the significance of PNI, GPS, and SIR markers as a prognostic factor for patients with colorectal neuroendocrine tumors (CR-NET), in various TNM stages. This study included 25 consecutive patients who were diagnosed with CR-NET and a control group consisting of 60 patients with newly diagnosed colorectal cancer (CRC). The study reported that CR-NET patients showed significantly higher serum levels of DA compared to CRC patients. The authors also concluded that a close relationship between the levels of DA and the inflammation and nutritional status of the CR-NET patients in this study.
Overall, the manuscript was good. However, a few major concerns/comments needed to be explained or modified.
- First time TNM was used in the abstract part so it should be in full form
- Line 67 needs a suitable reference
- Line 79 The reference style of citing in the manuscript was not matched as they can use [12-19], please check at a different place as well such as line 96, 101,102,107
- Line 164-165 what would be the possible reason for this
- Line 186 did the authors calculate the cutoff values or take from other research studies
- Line 272 LMR should be lymphocytes to monocytes ratio if I am right
- Line 285-286 It would be nice if the authors added the novelty of this study in the first paragraph of the discussion.
- Line 304 Please report previous studies shows that in tumor patients the DA was higher, please explain in a better way
- Line 326 So it means that in CR-NET patients the levels of NLR and PLR were higher, if yes explain your results with previous publications
- Please add figures or table numbers in the text of the discussion so that readers can easily follow up of your manuscript
- Please check the reference style as it did not match with MDPI journals
Author Response
Dear Reviewer,
Thank you very much for taking the time to analyze our manuscript, and for your kind appreciation and valuable suggestions.
All the type of recommended changes were performed in the body of our manuscript, with the Track Changes function activated.
Reviewer suggestions and comments
The authors in this study examined the presence and clinical application of serum DA, ST, NE, and EPI, in addition to determining the significance of PNI, GPS, and SIR markers as a prognostic factor for patients with colorectal neuroendocrine tumors (CR-NET), in various TNM stages. This study included 25 consecutive patients who were diagnosed with CR-NET and a control group consisting of 60 patients with newly diagnosed colorectal cancer (CRC). The study reported that CR-NET patients showed significantly higher serum levels of DA compared to CRC patients. The authors also concluded that a close relationship between the levels of DA and the inflammation and nutritional status of the CR-NET patients in this study.
Overall, the manuscript was good.
However, a few major concerns/comments needed to be explained or modified.
1. First time TNM was used in the abstract part so it should be in full form
- Revised
2. Line 67 needs a suitable reference
- Revised
3. Line 79 The reference style of citing in the manuscript was not matched as they can use [12-19], please check at a different place as well such as line 96, 101,102,107
- Revised
4. Line 164-165 what would be the possible reason for this
- We specified in text:
- This finding may suggest that patients with CR-NET have a considerably higher in-flammatory status compared to those with CRC, as also evidenced by the higher num-ber of LYMs; it is well recognized that LYMs are crucial to the host's immune response to prevent the growth and spread of tumors [48]. Patients in our study group also presented considerably higher serum values of ST, as shown above. Hanoun et al. [49], and Yoo et al. [50] showed that ST regulates inflammation by affecting the immune system. The role of ST in the gastrointestinal inflammatory response by activating immune cells to release inflammatory cytokines has been more clearly demonstrated in a number of animal studies [51,52,53]. Other studies have highlighted the presence of biodirectional neuroimmune interactions at the gut level in the regulation and consequences of intestinal inflammation, such as the central role that serotonin plays as a signaling molecule in triggering, intensifying, and combating inflammation [54,55]. These reports validate the gut's neuroimmune connections. The above may represent possible explanations, why patients with CR-NET showed significantly higher values of the CRP and inflammatory status compared to CRC patients.
5. Line 186 did the authors calculate the cutoff values or take from other research studies
- We calculated the cutoff values, ​​were not taken from other research studies. Moreover, we have specified in the text:
- Provided that 47.00 was the median value among the 25 CR-NET patients, and 48.00 for the 60 CRC patients, respectively, we used the median of PNI scores as classified criteria and were divided into two groups: Low PNI (<47.00, and <48.00, respectively) group and high PNI (≥ 47.00, and ≥48.00, respectively) group.
6. Line 272 LMR should be lymphocytes to monocytes ratio if I am right
- Revised in all situation
7. Line 285-286 It would be nice if the authors added the novelty of this study in the first paragraph of the discussion.
- We specified in text, us you suggested
8. Line 304 Please report previous studies shows that in tumor patients the DA was higher, please explain in a better way
- We specified in text, us you suggested
9. Line 326 So it means that in CR-NET patients the levels of NLR and PLR were higher, if yes explain your results with previous publications
- We discussed in the text between lines 359-383
10. Please add figures or table numbers in the text of the discussion so that readers can easily follow up of your manuscript
- We specified in text, us you suggested
11. Please check the reference style as it did not match with MDPI journals
- Revised in all situation
Reviewer 2 Report
Comments and Suggestions for Authors
The article has few shortcomings:
1. The study included only 25 patients diagnosed with colorectal neuroendocrine tumors (CR-NET) and 60 patients with colorectal cancer (CRC). This small sample size limits the generalizability of the findings and may not represent the broader population of patients with these conditions.
2. The authors acknowledge the absence of a pilot study and prior data referenced in the literature, which prevented them from determining the effect size and computing the sample size. This limitation affects the robustness and reliability of the study's conclusions.
3. The research was conducted at a single reference center, which may introduce bias and limit the applicability of the results to other settings or populations. Multicentric studies are generally preferred to enhance the external validity of the findings.
4. The study did not include patients with poorly differentiated (G3) tumors or those in TNM stage IV. This exclusion limits understanding the full spectrum of CR-NET and CRC, particularly in more advanced and aggressive cases.
5. The study was conducted over two years, and the authors mention constraints related to completing a PhD thesis. These limitations may have impacted the depth and breadth of the research.
6. The study did not account for potential confounding factors such as lifestyle, comorbidities, and treatment variations that could influence the levels of neurotransmitters and inflammatory markers. This oversight could affect the accuracy of the correlations observed.
7. While the study employed various statistical tests, the small sample size and the lack of a pilot study may have affected the power of the statistical analysis. The results might need to be more robust to draw definitive conclusions.
8. The study focused on a specific set of neurotransmitters and inflammatory markers. Other potentially relevant biomarkers were not investigated, which could provide a more comprehensive understanding of the disease mechanisms and prognostic factors.
9. Although the study was conducted by the Declaration of Helsinki and approved by the Ethics Committee, the small sample size and single-center nature raise questions about the ethical implications of generalizing the findings to a broader population without further validation.
While the study provides valuable insights into the correlation between neurotransmitters, nutritional and inflammatory markers, and TNM staging in CR-NET patients, its limitations necessitate caution in interpreting the results and highlight the need for further research with larger, multicentric cohorts.
Comments on the Quality of English Languageminor
Author Response
Dear Reviewer,
Thank you very much for taking the time to analyze our manuscript, and for your kind appreciation and valuable suggestions.
All the type of recommended changes were performed in the body of our manuscript, with the Track Changes function activated.
Comments and Suggestions for Authors
The article has a few shortcomings:
- The study included only 25 patients diagnosed with colorectal neuroendocrine tumors (CR-NET) and 60 patients with colorectal cancer (CRC). This small sample size limits the generalizability of the findings and may not represent the broader population of patients with these conditions.
- The authors acknowledge the absence of a pilot study and prior data referenced in the literature, which prevented them from determining the effect size and computing the sample size. This limitation affects the robustness and reliability of the study's conclusions.
- The research was conducted at a single reference center, which may introduce bias and limit the applicability of the results to other settings or populations. Multicentric studies are generally preferred to enhance the external validity of the findings.
- The study was conducted over two years, and the authors mention constraints related to completing a PhD thesis. These limitations may have impacted the depth and breadth of the research.
- While the study employed various statistical tests, the small sample size and the lack of a pilot study may have affected the power of the statistical analysis. The results might need to be more robust to draw definitive conclusions.
- Although the study was conducted by the Declaration of Helsinki and approved by the Ethics Committee, the small sample size and single-center nature raise questions about the ethical implications of generalizing the findings to a broader population without further validation.
* While the study provides valuable insights into the correlation between neurotransmitters, nutritional and inflammatory markers, and TNM staging in CR-NET patients, its limitations necessitate caution in interpreting the results and highlight the need for further research with larger, multicentric cohorts.
- Dear reviewer, regarding shortcomings 1,2,3,5,7,9 and * , with all due respect, you are only stating the limitations of our study, which the authors have assumed at the end of the discussion section
- The study did not include patients with poorly differentiated (G3) tumors or those in TNM stage IV. This exclusion limits understanding the full spectrum of CR-NET and CRC, particularly in more advanced and aggressive cases.
- We specified in the text: After the HP examination, we noticed that in the CR-NET group patients, we did not identify poorly differentiated (G3 NET) tumors. Additionally, we found no metas-tases in any patient, indicating the absence of TNM stage IV cases. Based on the obtained results, we excluded 43 patients with poorly differentiated (G3) tumors and 22 patients with TNM stage IV diagnoses from the CRC group.
- Since in the group of 25 CR-NET patients we did not diagnose any patients with poorly differentiated tumors (G3) or TNM stage IV patients, we made the decision not to include patients with G3 or TNM stage IV tumors from the CRC group in order not to influence the statistical results.
- The study did not account for potential confounding factors such as lifestyle, comorbidities, and treatment variations that could influence the levels of neurotransmitters and inflammatory markers. This oversight could affect the accuracy of the correlations observed.
- Out of the 25 patients included in the study, 7 were alcohol consumers, 6 were smokers, and 5 had stage I or II hypertension. We do not exclude them from such a small existing lot.
- The study focused on a specific set of neurotransmitters and inflammatory markers. Other potentially relevant biomarkers were not investigated, which could provide a more comprehensive understanding of the disease mechanisms and prognostic factors.
- In our study, we did not aim to investigate other potentially relevant biomarkers.
- In the future, we propose to continue the study and enroll patients from other reference centers in Romania to see if the results obtained with this group of patients will be replicated.
Round 2
Reviewer 2 Report
Comments and Suggestions for Authors
I am satisfied with the revisions.
thank you.